# Discharge policies and care practices for children with suspected sepsis: A health facility scan at a nationally representative sample of hospitals and health centres in Uganda

Jessica Trawin[1], Martina Knappett[1], Clare Komugisha[2], Savio Mwaka[2], Ezrah Bamwesigye[2], Collins Agaba[2], David Walugembe[1], Jesca Nsungwa Sabiiti[3], J. Mark Ansermino[1,4], Niranjan Kissoon[1,5], Nathan Kenya Mugisha[2], Matthew O. Wiens[1,2,4]*

1 Institute for Global Health, British Columbia Children's Hospital and British Columbia Women's Hospital+Health Centre, Vancouver, Canada, 2 World Alliance for Lung and Intensive Care Medicine, Kampala, Uganda, 3 Department of Reproductive and Child Health, Ministry of Health, Kampala, Uganda, 4 Department of Anesthesiology, Pharmacology and Therapeutics, University of British Columbia, Vancouver, Canada, 5 Department of Pediatrics, University of British Columbia, Vancouver, Canada

* matthew.wiens@bcchr.ca

## Abstract

Under-five children in low- and middle-income countries remain at high risk of death after hospital discharge. However, few studies have systematically assessed discharge processes or facility readiness to support safe transitions of care. This study aimed to assess health facility readiness to provide pediatric discharge care for children under five years of age and current discharge practices in a nationally representative sample of health facilities in Uganda. A cross-sectional health facility scan was conducted between October 2020 and May 2021 at 36 facilities providing inpatient pediatric care in Uganda. Primary outcomes included: (1) facility readiness for pediatric discharge, defined as availability of infrastructure, technology, forms/job aids, and equipment; and (2) observed discharge care practices, including caregiver counselling, provision of take-home materials, post-discharge risk assessment, referrals, and clinical assessments on the day of discharge. Secondary outcomes included discrepancies between reported versus observed discharge care practices, discharge relevant admission practices, as well as caregiver and health worker satisfaction. Thiry-six health facilities were enrolled and 180 pediatric discharge observations, 180 caregivers, and 180 health workers were assessed. Hospitals had higher readiness scores in infrastructure (p < 0.001), technology (p = 0.006), and equipment (p = 0.021) than health centres. Hospitals also performed better in the provision of discharge risk assessment, clinical assessment on the day of discharge, follow-up, and provision of take-home materials. In contrast, health centres more consistently provided discharge counselling (p = 0.021) and had higher counselling topic scores (p < 0.001). Overall, 82.8% of discharges included a clinical assessment, 31.1% included a

**Data availability statement:** Study materials (protocol, consent forms, data collection tools, and metadata) are publicly available through the Pediatric Sepsis Data CoLaboratory's (Sepsis CoLab) Dataverse on Borealis, the Canadian Dataverse Repository. Due to the sensitive nature of clinical data and the potential risk for re-identification of research participants, the de-identified dataset is available through moderated access on the Sepsis CoLab Dataverse. Access to these data will be granted on a case-by-case basis following approval from the authors and the Data Governance Committees. Any requests to access these data can be made to the data governance committee at the following email address: sepsiscolab@bcchr.ca.

**Funding:** MOW and NKM received funding (#TTS-1809-19395) for this study from Grand Challenges Canada (GCC) through WALIMU in Uganda and the BC Children's Hospital Foundation (BCCHF) through the Institute for Global Health at BC Children's Hospital in Canada. The funders had no role in study design, data collection and analysis, decision to publish or preparation of the manuscript.

**Competing interests:** The authors have declared that no competing interests exist.

follow-up referral, and 26.1% included a risk assessment. Observed practices often diverged from reported procedures. These findings identified several priority areas for quality improvement in both resource availability and discharge care delivery in all settings. Standardizing discharge policies and tools may strengthen discharge care and may be used as a guide to inform national-level pediatric discharge policies.

## Introduction

In 2021, over five million children under the age of five died globally, with sub-Saharan Africa (SSA) accounting for 53% of these deaths [1]. Most of these deaths are due to infectious diseases, with sepsis being the leading immediate cause of death in SSA and South Asia [2–4]. Uganda has one of the highest under-5 mortality rates in SSA at 58.4 deaths per 1,000 children [5]. An important, but neglected, aspect of child mortality are deaths occurring during the post-discharge period which often exceed the burden of in-hospital deaths [6–9]. Thus, addressing post-discharge deaths would contribute to the Sustainable Development Goals (SDGs) relating to under-5 mortality.

Most post-discharge deaths occur in the community and hence require a health systems approach to address existing gaps in pediatric discharge procedures across the spectrum of care [9–11]. Discharge is often considered a complex and error-prone process which has been linked to poor post-discharge outcomes [11,12]. Little work has been done in low- and middle-income countries (LMICs) to understand existing discharge practices and care processes. Standardized policy-driven procedures can improve the safety and efficiency of discharge care in both high and poor resource settings [13–15]; however, pediatric discharge processes in Uganda are largely based on informal hospital-specific protocols or clinician opinions [16].

The World Health Organization's (WHO) Guidelines for the Management of Common Childhood Illnesses provide key elements to effective hospital discharge: i) discharge timing; ii) comprehensive post-discharge counselling; iii) immunization and record keeping; iv) post-discharge follow-up referrals; and v) assisting families with special support [17]. This study aligns with WHO guidelines by evaluating how well health facilities in Uganda implement these core elements, providing a health systems-based assessment of existing resources and practices in pediatric discharge. Specifically, our study aimed to assess health facility readiness to provide pediatric discharge care for children under five years of age and current discharge practices in a nationally representative sample of health facilities in Uganda.

## Methods

### Study design and setting

This cross-sectional, survey-based study was conducted between 28 October 2020 and 5 May 2021 at health facilities providing in-patient pediatric care across Uganda. We purposively selected 36 health centres (Level III and IV) and hospitals from 20 districts to comprise a sample that was nationally representative of the Ugandan

health system and the socio-demographic diversity of the local population (Table A in S1 Text). In Uganda, health centres range from level I to level IV, though only level III and IV admit patients. Health centre level III and IV facilities constitute approximately 90% of the nearly 2000 admitting facilities in Uganda. Hospitals, which offer more advanced and specialist services, represent about 10% of admitting facilities [18,19]. Government ownership represents 45% of these facilities, with 40% being private for profit and 15% private-not-for-profit (PNFP). In Uganda, public facilities offer health services and medications at no cost, whereas fees vary across private facilities [18]. Private-for-profit health facilities were excluded due to differing funding models and patient populations, which are less representative of the broader public and not-for-profit sectors serving most Ugandan children.

## Facility scan tool

This study used a 5-survey Facility Scan informed by the Pediatric Sepsis Data CoLaboratory's (Sepsis CoLab) Environmental Scan to evaluate discharge practices both at the facility level and at the patient level [20,21,22]. Two of these surveys focus on facilities, assessing (1) facility resources (survey #1) and (2) facility technological preparedness (survey #2). The other three surveys focus on individuals (patients, caregivers and health workers) and include (1) a scan to assess the quality and safety of care through the observation of a health workers caring for children (survey #3), (2) a caregiver satisfaction questionnaire to assess perceptions of the care received (survey #4) and (3) a health worker satisfaction questionnaire to assess health worker perceptions of the care provided (survey #5). Within each of the five surveys, viables were divided into several domains. A detailed description of the 5-survey Facility Scan is provided in the supplement (Table A in S1 Text).

## Facility recruitment

A local study team member initially contacted District Health Offices (DHOs) via email and arranged in-person meetings for facility recruitment. Hospital administrators who agreed to participate were enrolled and contacted by phone one week prior to the initiation of study activities at their respective sites.

## Participant recruitment

Convenience sampling was used to identify a total sample size of 180 admitted children (Observational Scan), 180 caregivers (Caregiver Satisfaction Questionnaire), and 180 health workers (Health Worker Satisfaction Questionnaire), five of each from each of the 36 sites. For the observational scan, children were observed if they were admitted due to suspected sepsis and were under 5 years of age. For the Caregiver Satisfaction Questionnaire, a research assistant approached caregivers for consent and enrollment at the time of discharge if they were the primary caregiver of a child under the age of five who was discharged following inpatient treatment for suspected sepsis and was actively involved in their child's discharge experience or knowledgeable about their child's discharge experience from having interactions with health workers or their child's other caregivers. Caregiver Satisfaction Questionnaire participants were not necessarily the same as those included in the Observational Scan. For the Health Worker Satisfaction Questionnaire, health workers were purposively selected to participate based on their level of experience and through consultations with site leads. Health workers who had been working in the pediatric departments at participating health facilities for at least two months were eligible to be included.

## Study procedures

The two facility surveys comprising of the Environmental Scan (Survey 1) and Technology Readiness Scans (Survey 2) were collected by trained research assistants who worked both independently and at times through consultations with a hospital administrator using a series of checklists and closed-ended questions. The Observational Scan (Survey 3)

involved a research assistant observing health workers throughout the admission and discharge process for patients receiving in-hospital treatment for severe infection, comprising a single observation per patient. Observations were recorded using a closed-ended survey. The Caregiver Satisfaction Questionnaire (Survey 4) and Health Worker Satisfaction Questionnaires (Survey 5) were administered by a research assistant and data were collected using a series of closed-ended questions and Likert scales. Data were collected offline using the Research Electronic Data Capture (RED-Cap) Mobile Application [23] and uploaded to the REDCap server after completion of the Facility Scan at each site.

## Study outcomes

This study has two co-primary outcomes, one measured at the facility level, and one measured at the individual level. First, we sought to assess facility readiness for discharge, defined as resource availability in four individual domains: i) infrastructure; ii) technology; iii) forms & job aids; iv) equipment. Second, we sought to assess five key discharge care practices: i) counselling on post-discharge care; ii) provision of take-home materials; iii) post-discharge risk assessment (subjectively defined); iv) referrals and v) conducting a clinical assessment on the day of discharge. Secondary outcomes included key practices that were observed to be performed versus key practices reported as standard of practice at the facility level. The practice of conducting a clinical assessment on the day of discharge was not compared in this manner as it was assumed all facilities would report this as being necessary. Secondary outcomes also included admission features relevant to discharge, as well as caregiver and health worker satisfaction with discharge care.

## Survey and statistical analysis

The survey analysis was descriptive in nature. The frequency and distribution of each variable was calculated and reported in a stratified manned by both service delivery level (grouped as either hospital or health centre) and ownership type (public or PNFP). The two facility surveys (Environmental Scan and Technology Readiness Scan) were grouped and reported across four domains: Infrastructure (7 items), Technology (7 items), Forms & Job Aids (6 items) and Equipment (8 items). Each domain score was derived by calculating the proportion of available items. The domain scores reflect the extent of facility readiness in terms of infrastructure, technology, and resources for discharge care (e.g., discharge forms, counselling guides, etc.), and constituted the first co-primary outcome. From the Observational Scan (Survey 3), care practice scores were calculated. These were grouped across seven domains: medical history taking (7 items), socio-economic history taking (7 items), physical measurements (8 items), clinician observed measurements (14 items), inpatient consultation (6 items), discharge forms (14 items), and counselling topic (11 items). Like the facility surveys, domain scores were calculated as the proportion of observed items. From these seven domains, the five key discharge care practices which constituted the second co-primary outcome were individually reported. Each of these five variables (post-discharge counselling, provision of take-home materials, post-discharge risk assessment, post-discharge referrals, and clinical assessment on the day of discharge) were binary variables, assessed as either present or absent. For the purposes of the primary outcome, no consideration was given to the content of these practices (ex. counselling topics, method of risk assessment, etc).

To assess the secondary outcome of reported versus observed discharge practices, we compared facility-level responses from the Facility Scan (Survey 1) indicating whether each key discharge activity was reported to be routinely performed, with actual performance observed in the 5 observations for that site according to the Observational Scan (Survey 3). Observations were categorized into four groups: observed and reported, observed but not reported, not observed but reported, and not observed and not reported. Both readiness and care practice domain scores were stratified by service delivery level and ownership type and summarized by median and interquartile ranges (IQR). Chi-Squared (or Fisher's Exact when the sample size was small) and Mann-Whitney U tests were performed to determine differences between service delivery levels and ownership type, with $p$-values lower than 0.05 considered as statistically significant.

Microsoft Excel (Microsoft Corporation, Redmond, WA), R version 4.2.2 (R Foundation for Statistical Computing, Vienna, Austria), and RStudio version 2022.12.0 (RStudio, Boston, MA) were used for analysis.

### Missing data

Missing data were removed from the analysis. Missing data was minimal, though in once instance 5 baseline data-points (3 from the Observational scan, 2 from the Caregiver Satisfaction Questionnaire) from 7 sites were lost from a single study device during an upload error. This resulted in 145 observations rather than 180 observations for these 5 baseline variables. These baseline variables related to descriptive features of patients (sex, diagnosis, discharge status) and caregivers (length of stay of their child, travel time to facility), but were not part of the formal survey responses, which were nearly complete across all participants.

### Ethics approval and consent to participate

Ethics approval was obtained from the University of British Columbia/Children's and Women's Health Centre of British Columbia Research Ethics Board (UBC C&W REB # H20-02519) and the Makerere University School of Public Health, Higher Degrees, Research and Ethics Committee (HDREC # 851). The study was registered with the Uganda National Council for Science and Technology (HS928ES). A waiver of consent was obtained from both institutional review boards for facility participation in the Environmental, Technology Readiness, and Observational Scans. Written informed consent, obtained through either a signature or thumbprint, was obtained from individuals participating in the Caregiver and Health Worker Satisfaction Questionnaires. Consent forms were available in commonly spoken local languages in each region as well as English. Administrative clearance was provided by District Health Offices prior to initiating data collection.

## Results

For the facility surveys, we invited 37 facilities to participate and enrolled 36 into the study, including 23 hospitals (17 public and 6 PNFP) and 13 health centres (11 public and 2 PNFP) (Fig 1). The observational scan was completed in 180 pediatric patients admitted and discharged following their treatment for suspected sepsis, though baseline features were lost for 35 of these observations. Among the 145 with baseline data, it was observed that 98% (142 of 145) of patient observations were routinely discharged. The most common diagnoses were malaria (39.3%, n = 57) and pneumonia (22.8%, n = 33) with the remaining cases (37.9%, n = 57) grouped as 'other or multiple infections' due to low frequency. A total of 180 caregivers were interviewed and completed the caregiver satisfaction survey. Similar to the observational scan, some baseline data were missing in 36 cases. Among those in whom baseline data was captured, their child's median length of stay was three days (IQR: 2–4), and most caregivers (57.2%, n = 83) lived within 1 hour of the health facility where their child was admitted. Among the 180 health workers interviewed, the median length of employment at the facility was two years (IQR: 1–5). Most health workers (69.4%, n = 152) were nursing or allied health staff, with the remaining being physicians (23.9%, n = 43) or clinical officers (6.7%, n = 12).

### Resource availability

Across all four domains of the primary outcome measured at the facility level it was observed that hospitals had significantly higher domain scores for Infrastructure (p < 0.001), technology (p = 0.06) and equipment (p = 0.021) (Fig 2, Table 1, Fig A in S1 Text). There were no observed overall differences between ownership types (public vs PNFP) across any of these four domains. All facilities had on-site pharmacies, with 80.6% (n = 29) providing free medications; all seven facilities that charged patients for medications were PNFP. Nearly all facilities reported experiencing stock outs of essential drugs lasting over 24 hours, with stock outs more frequent in public facilities than PNFP (p = 0.028). About one-third of facilities had referral clinics for follow-up, though compared to health centres, hospitals were significantly more likely to have such

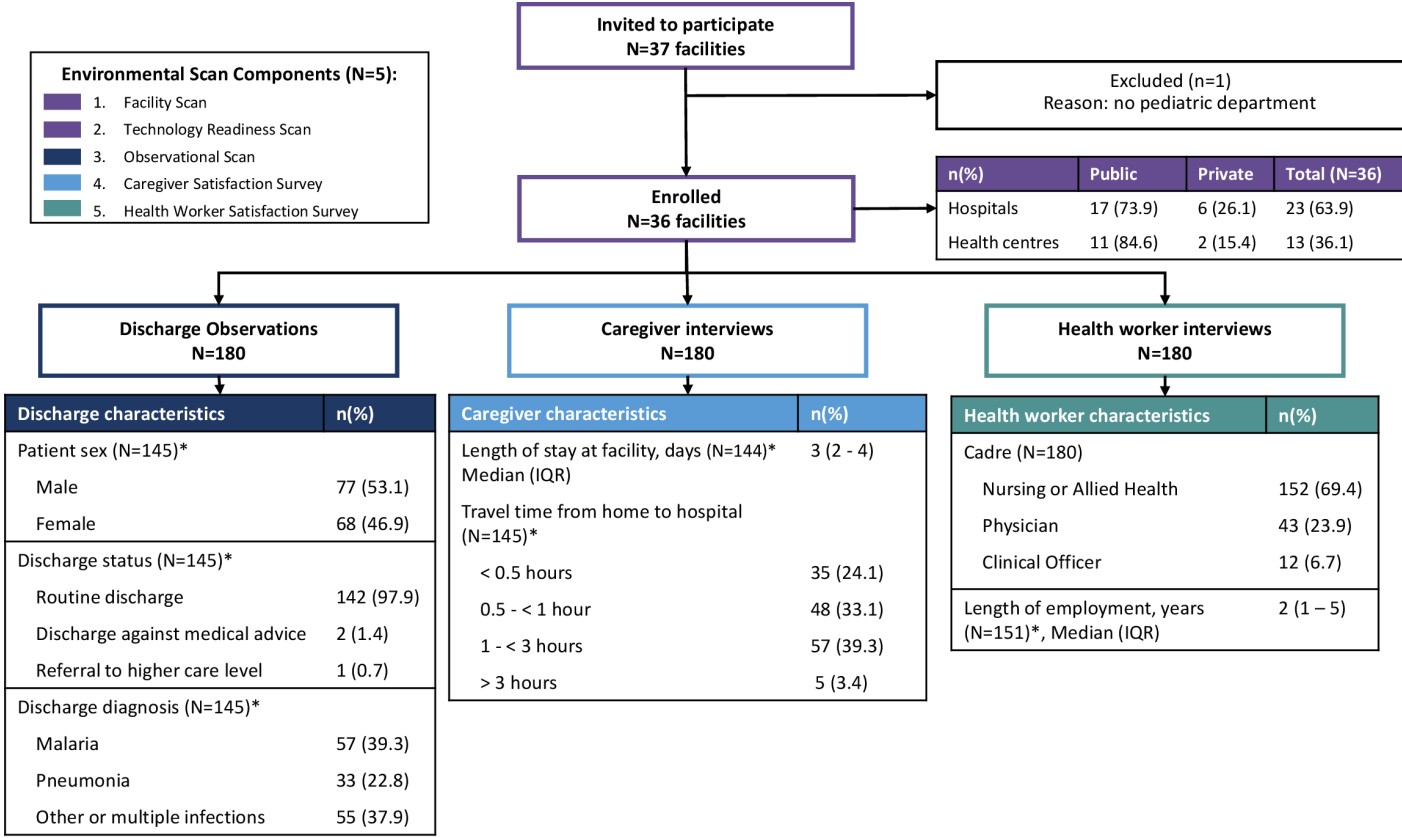

**Fig 1. Facility enrollment and participant characteristics.** * Missing baseline data due to uploding error (missingness of baseline data did not impact survey missingness).

referral clinics ($p = 0.011$). Hospitals had a significantly higher Technology domain score (7 items), with a median of 3 (IQR: 2–4) compared to health centres who had a median score of 1 (IQR: 0–2) ($p = 0.006$). Hospitals were also significantly more likely to have Electronic Medical Record (EMR) systems (87.0%; n = 20) than health centres (30.8%; n = 4) ($p = 0.001$).

In the equipment domain, hospitals fared better than health centers. Hospitals also had a higher median Equipment domain score. Out of 8 items, they had a median of 6 (IQR: 5–7) compared to health centres, who had a median of 5 (IQR: 4–5) ($p = 0.021$). Notably, 73.9% (n = 17) of hospitals had pulse oximeters available compared to only 7.7% (n = 1) of health centres ($p < 0.001$). The Forms & Job Aids domain score (7 items) was similar between health centres. Hospitals and PNFP facilities were more likely to have discharge forms compared to health centres ($p = 0.030$) and public facilities ($p = 0.032$). Only one facility (2.8%) had a discharge policy and no facilities had counselling guides available.

## Care practices

**Key discharge practices.** For the primary outcome related to key discharge care practices, discharge clinical assessments and counselling on post-discharge care were the most frequently completed practices, being observed in 149 (83%) and 113 (63%) of cases, respectively (**Table 2**). Scheduling follow-up referrals (31%), conducting post-discharge risk assessments (26%), and the provision of take-home materials (25%), and were observed to occur much

Axes represent average proportion of items present across all facilities.

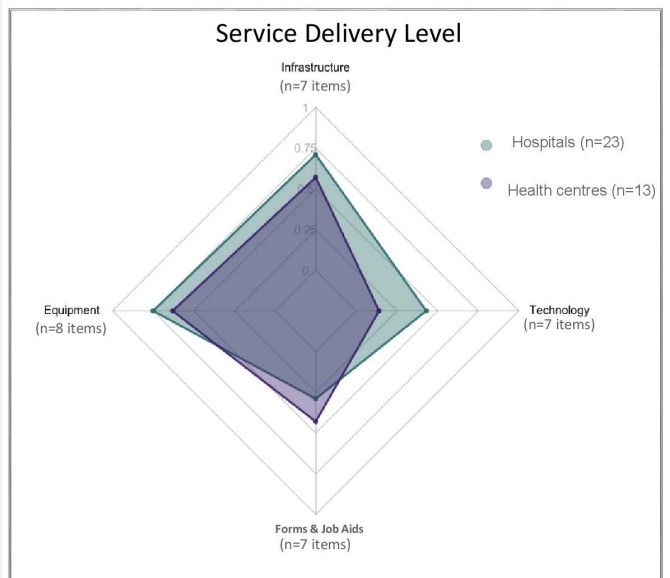

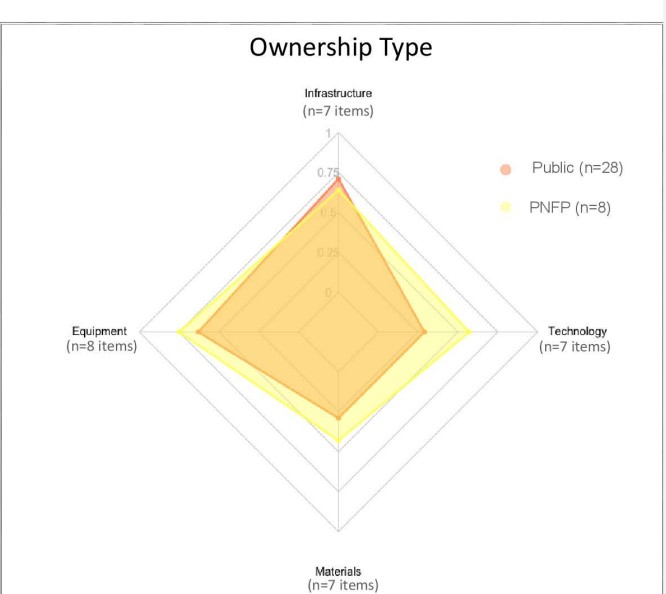

**Fig 2. Polar graph of resource availability by service delivery level and ownership type (N = 36 facilities).** Axes represent average proportion of items present across all facilities.

less frequently. We observed important differences between health centres and hospitals across the five key discharge practices.

With regards to discharge counselling, about two-thirds of sites provided at least some degree of counselling on post-discharge care, though it occurred more frequently at health centres than hospitals ($p = 0.021$). Furthermore, where health workers provided discharge counselling, health centres achieved a significantly higher Counselling Topic score (11 items), with a median of 8 topics discussed (IQR: 6–10) compared to hospitals with a median of 5 (IQR: 4–6) ($p < 0.001$) (Table 2). Medication management/dosing, discharge diagnosis, and medication type/purpose were the most frequently discussed topics across all case observations, while medication side effects, immunization, and recovery were the least frequently discussed.

Compared to health centre health workers, it was observed that hospital health workers were significantly more likely to provide take-home materials ($p < 0.001$). Take-home materials consisted exclusively of discharge forms, rather than counselling or caregiver guides. Health workers provided discharge forms to caregivers in 35.7% (n = 41) of hospital cases but in only 6.2% (n = 4) of health centre cases (Table 2). However, discharge forms were more complete when provided at health centres (p = 0.04). Hospitals also were observed to consider post-discharge risk more often than at health centres (40% vs 1.5%, p < 0.001). All risk assessments relied on the health worker's opinion rather than an objective risk assessment tool or algorithm. Hospitals were also observed to schedule follow-up referrals more often (39% vs 19%, $p = 0.021$) as well as more frequently conduct clinical assessments on the day of discharge (97% vs 59%, p < 0.001).

We identified discrepancies between reported discharge procedures at the facility-level compared to observed care practices across four key discharge practices (Fig 3, Table B in S1 Text). Health workers did not conduct risk assessments for post-discharge mortality in 27.8% of hospital case observations and 60.0% of health centre cases, despite facilities reporting this as a key discharge procedure. Additionally, health workers did not provide discharge counselling in 36.5% of hospital cases and 10.8% of health centre cases, or take-home materials to caregivers in 64.3% of hospital cases and 93.8% of health centre cases, though these were reported as standard step in the discharge process. Lastly, health

**Table 1. Resource availability by service delivery level and ownership type, n = 36 facilities.**

| Domain | Hospitals (n = 23) | | Health Centres (n = 13) | | | PNFP (n = 8) | | Public (n = 28) | | | All (n = 36) | |
|---|---|---|---|---|---|---|---|---|---|---|---|---|
| | n | % | n | % | p-value | n | % | n | % | p-value | n | % |
| **Infrastructure** | | | | | | | | | | | | |
| Children are kept in a separate ward or separate area of a ward from adult patients | 23 | 100.0% | 3 | 23.1% | <0.001* | 6 | 75.0% | 20 | 71.4% | 1 | 26 | 72.2% |
| Mothers have access to running water and to an appropriate space to wash | 23 | 100.0% | 12 | 92.3% | 0.361 | 8 | 100.0% | 27 | 96.4% | 1 | 35 | 97.2% |
| Facility has referral clinics that follow-up children post-discharge | 13 | 56.5% | 1 | 7.7% | 0.011 | 4 | 50.0% | 10 | 35.7% | 0.683 | 14 | 38.9% |
| Facility has on-site pharmacy for patients to fill prescriptions post-discharge | 23 | 100.0% | 13 | 100.0% | NA | 8 | 100.0% | 28 | 100.0% | NA | 36 | 100.0% |
| On-site pharmacy provides essential drugs to patients/caretakers for free | 18 | 78.3% | 11 | 84.6% | 1 | 1 | 12.5% | 28 | 100.0% | <0.001* | 29 | 80.6% |
| No essential drugs for treating children with infection at the pharmacy are expired | 19 | 82.6% | 11 | 84.6% | 1 | 7 | 87.5% | 23 | 82.1% | 1 | 30 | 83.3% |
| Facility has never had a stock-out for more than 24 hours of essential drugs | 4 | 17.4% | 0 | 0.0% | 0.274 | 3 | 37.5% | 1 | 3.6% | 0.028* | 4 | 11.1% |
| 7-item domain index score, Median (IQR) | 5 | 5 - 6 | 4 | 4–4 | <0.001* | 4.5 | 3.75–5.25 | 5 | 4–6 | 0.496 | 5 | 4–6 |
| **Technology** | | | | | | | | | | | | |
| Triage process follows a standard algorithm/process | 11 | 47.8% | 1 | 7.7% | 0.605 | 4 | 50.0% | 8 | 28.6% | 0.358 | 12 | 33.3% |
| EMR system use | 20 | 87.0% | 4 | 30.8% | 0.001* | 6 | 75.0% | 18 | 64.3% | 0.691 | 24 | 66.7% |
| Local network and server use | 15 | 65.2% | 1 | 7.7% | 0.003* | 5 | 62.5% | 11 | 39.3% | 0.422 | 16 | 44.4% |
| Access to IT personnel | 13 | 56.5% | 5 | 38.5% | 0.488 | 5 | 62.5% | 13 | 46.4% | 0.691 | 18 | 50.0% |
| Computer availability | 10 | 43.5% | 0 | 0.0% | 0.006* | 3 | 37.5% | 7 | 25.0% | 0.658 | 10 | 27.8% |
| Use of tools/algorithm for discharge vulnerability | 0 | 0.0% | 0 | 0.0% | NA | 0 | 0.0% | 0 | 0.0% | NA | 0 | 0.0% |
| Mobile device availability | 0 | 0.0% | 7 | 53.8% | <0.001* | 0 | 0.0% | 7 | 25.0% | 0.309 | 7 | 19.4% |
| 7-item domain index score, Median (IQR) | 3 | 2 - 4 | 1 | 0–2 | 0.006* | 4 | 0.75–4.25 | 2 | 1–3 | 0.440 | 2 | 1–4 |
| **Forms & Job Aids** | | | | | | | | | | | | |
| Stock cards/drug log book | 20 | 87.0% | 12[a] | 100.0% | 0.536 | 7 | 87.5% | 25[b] | 92.6% | 0.553 | 32[c] | 91.4% |
| Discharge forms | 18 | 78.3% | 5 | 38.5% | 0.030* | 8 | 100.0% | 15 | 53.6% | 0.032* | 23 | 63.9% |
| IMCI chart booklet | 11 | 47.8% | 7 | 53.8% | 1 | 2 | 25.0% | 16 | 57.1% | 0.229 | 18 | 50.0% |
| Child vaccination cards | 2 | 8.7% | 12 | 92.3% | <0.001* | 3 | 37.5% | 11 | 39.3% | 1 | 14 | 38.9% |
| Follow-up referral forms | 1 | 4.3% | 0 | 0.0% | 1 | 1 | 12.5% | 0 | 0.0% | 0.222 | 1 | 2.8% |
| Counselling guide | 0 | 0.0% | 0 | 0.0% | NA | 0 | 0.0% | 0 | 0.0% | NA | 0 | 0.0% |
| Discharge policy | 1 | 4.3% | 0 | 0.0% | 1 | 0 | 0.0% | 1 | 3.6% | 1 | 1 | 2.8% |
| 7-item domain index score, Median (IQR) | 2 | 2 - 3 | 3 | 2–3 | 0.096 | 3 | 2–3 | 2 | 2–3 | 0.603 | 2.5 | 2–3 |
| **Equipment** | | | | | | | | | | | | |
| Thermometer | 21 | 91.3% | 13 | 100.0% | 0.525 | 8 | 100.0% | 26 | 92.9% | 1 | 34 | 94.4% |
| Weigh machine/scales for children | 21 | 91.3% | 11 | 84.6% | 0.609 | 6 | 75.0% | 26 | 92.9% | 0.208 | 32 | 88.9% |
| MUAC tape/measuring tape | 21 | 91.3% | 10 | 76.9% | 0.328 | 5 | 62.5% | 26 | 92.9% | 0.062 | 31 | 86.1% |
| Measuring board/ruler | 18 | 78.3% | 9 | 69.2% | 0.693 | 5 | 62.5% | 22 | 78.6% | 0.384 | 27 | 75.0% |
| Pulse oximeter | 17 | 73.9% | 1 | 7.7% | <0.001* | 6 | 75.0% | 12 | 42.9% | 0.229 | 18 | 50.0% |
| Height measuring scale/ruler | 13 | 56.5% | 10 | 76.9% | 0.292 | 4 | 50.0% | 19 | 67.9% | 0.422 | 23 | 63.9% |
| Blood pressure machine | 11 | 47.8% | 2 | 15.4% | 0.075 | 4 | 50.0% | 9 | 32.1% | 0.422 | 13 | 36.1% |

*(Continued)*

**Table 1.** (Continued)

| Domain | Hospitals (n = 23) | | Health Centres (n = 13) | | | PNFP (n = 8) | | Public (n = 28) | | | All (n = 36) | |
|---|---|---|---|---|---|---|---|---|---|---|---|---|
| | n | % | n | % | p-value | n | % | n | % | p-value | n | % |
| Pediatric stethoscope | 7 | 30.4% | 1 | 7.7% | 0.213 | 3 | 37.5% | 5 | 17.9% | 0.338 | 8 | 22.2% |
| 8-item domain index score, Median (IQR) | 6 | 5 - 7 | 5 | 4–5 | 0.021* | 6 | 2.75–7 | 5 | 5–6 | 0.683 | 5 | 4.75–6 |

Missing Data: [a]n=12; [b]n=27; [c]n=34; *Statistically significant p-value <0.05

workers did not schedule follow-up referrals in 33.0% of hospital cases and 7.8% of health centre cases, as per facility standards. Notably, health workers independently scheduled follow-up referrals in 18.8% of health centre cases, even though this was not reported as a discharge practice at their facility.

**Discharge relevant practices observed at admission.** At admission, facilities captured a median of 4 (IQR 3–6) items (out of 7) in the medical history domain, with health centres scoring slightly higher than hospitals ($p < 0.05$) (Fig S2, Table C in S1 Text). Health workers in general did not commonly perform and document socio-economic history taking, though health centres also had significantly higher scores in this 7-item domain ($p < 0.001$). Health centre staff were more likely than hospital staff to ask and document bed net use, boiling/disinfecting drinking water, distance from hospital, pit/latrine access, and maternal education.

Both hospitals and health centres infrequently performed and documented Physical Measurements (8 items), with equal median domain scores of 3 (IQR: 2–4, $p = 0.201$) (Fig B in S1 Text, Table C in S1 Text). Health workers assessed and documented weight in most cases (88.3%, n = 159). Other vital signs such as oxygen saturation (SpO$_2$) were measured and documented much less frequently (21.7%, n = 39), although no measurements occurred at health centres due to the absence of pulse oximeters. Both service delivery levels had low Clinician Observed Measurement scores (14 items), with a median of 3 items observed (IQR 0–7), though hospitals scored significantly higher ($p < 0.001$).

Of the 6-item in-patient consultation topics, the median that were discussed and documented was 1 (IQR 0–3), with health centres performing significantly better than hospitals ($p < 0.001$) (Fig B in S1 Text, Table C in S1 Text). The most common topics discussed were the type of patient illness, patient management and estimated length of stay. However, they discussed potential challenges caregivers may face during hospital stay, such as the cost of food and prescriptions, in only 5 (2.8%) of all cases.

**Caregiver satisfaction.** Caregivers at health centres reported significantly higher satisfaction with discharge preparation (n = 47, 72.3%) than those at hospitals (n = 58, 50.4%) ($p = 0.004$) (Table D in S1 Text). Over half of caregivers felt that they were involved in discharge decisions as much as they wanted to be (n = 97, 53.9%) and most caregivers (n = 147, 81.7%) found the timing of discharge suitable. Among the 59 (32.8%) caregivers referred for follow-up care, most (n = 55, 93.2%) were referred to their preferred facility.

**Health worker satisfaction.** Compared to hospitals, health centre health workers reported significantly higher satisfaction with the overall discharge process ($p = 0.010$), perceived peer satisfaction with the discharge process ($p = 0.010$), staff availability for discharging patients in general ($p = 0.008$), and staff availability for discharging patients on weekends ($p < 0.001$) (Table D in S1 Text). Whereas hospital health workers reported significantly higher satisfaction with the availability of support or supervision when discharging a child compared to those at health centres ($p = 0.040$). Overall, 45% (n = 81) of health workers were "always" or "often" pleased with the discharge process at their facility and 52.2% (n = 94) rated the discharge education provided at their facility as "good" or "satisfactory."

**Table 2. Comparison of observed discharge practices between hospitals and health centres.**

| Key discharge practice | Hospitals (n = 115), n (%) | Health Centres (n = 65), n (%) | All (N = 180), n (%) | p-value |
|---|---|---|---|---|
| Counselling caregiver on post-discharge care | 65 (56.5) | 48 (73.8) | 113 (62.8) | 0.021* |
| Providing take-home materials | 41 (35.7) | 4 (6.2) | 45 (25.0) | <0.001* |
| Conducting post-discharge risk assessment | 46 (40.0) | 1 (1.5) | 47 (26.1) | <0.001* |
| Scheduling follow-up referrals | 44 (38.3) | 12 (18.8)[a] | 56 (31.1)[b] | 0.007* |
| Conducting clinical assessments on day of discharge | 111 (96.5) | 38 (58.5) | 149 (82.8) | <0.001* |
| **Discharge counselling topics observed** | **Hospitals (n = 65), n (%)** | **Health Centres (n = 48), n (%)** | **All (N = 113), n (%)** | **p-value** |
| Medication side effects | 1 (1.6)[a] | 16 (33.3) | 17 (15.2)[c] | <0.001* |
| Immunization information | 15 (23.8)[d] | 14 (29.8)[d] | 29 (26.4)[e] | 0.628 |
| Recovery | 20 (31.3)[a] | 18 (37.5) | 38 (33.9)[c] | 0.624 |
| Warning signs | 14 (21.9)[a] | 37 (77.1) | 51 (45.5) | <0.001* |
| What to do if warning signs appear | 15 (23.4)[a] | 37 (77.1) | 52 (46.4)[c] | <0.001* |
| Hygiene information | 30 (46.9)[a] | 42 (89.4)[e] | 72 (64.9)[c] | <0.001* |
| Nutrition information | 39 (60.9)[a] | 37 (78.7)[c] | 76 (68.5)[c] | 0.074 |
| Mosquito net use | 39 (60.0) | 39 (81.3) | 78 (69.0) | 0.016* |
| Medication (type, purpose, etc.) | 40 (61.5) | 42 (87.5) | 82 (72.6) | 0.002* |
| Discharge diagnosis information | 47 (73.4)[a] | 46 (95.8) | 93 (83.0)[c] | 0.002* |
| Medication management/dosing | 51 (78.5) | 46 (97.9)[d] | 97 (86.6)[c] | 0.003* |
| Median (IQR) 11 item counselling topic score | 5 (3– 6) | 8 (6– 10) | 6 (4 – 8) | <0.001* |
| **Take-home materials provided at discharge** | **Hospitals (n = 115), n (%)** | **Health Centres (n = 65), n (%)** | **All (N = 180), n (%)** | **p-value** |
| Discharge form | 41 (35.7) | 4 (6.2) | 45 (25.0) | <0.001* |
| Caregiver's guide | 0 (0) | 0 (0) | 0 (0) | NA |
| Counselling guide | 0 (0) | 0 (0) | 0 (0) | NA |
| **Discharge form information** | **Hospitals (n = 40)[f], n (%)** | **Health Centres (n = 4), n (%)** | **All (N = 44), n (%)** | **p-value** |
| Name of patient | 40 (100.0) | 4 (100.0) | 44 (100.0) | – |
| Age of patient | 40 (100.0) | 4 (100.0) | 44 (100.0) | – |
| Date of admission | 40 (100.0) | 4 (100.0) | 44 (100.0) | – |
| Discharge medication | 40 (100.0) | 4 (100.0) | 44 (100.0) | – |
| Date of discharge | 36 (90.0) | 4 (100.0) | 40 (90.9) | 1 |
| Discharge diagnosis | 35 (87.5) | 4 (100.0) | 39 (88.6) | 1 |
| Treatment received in hospital | 32 (80.0) | 4 (100.0) | 36 (81.8) | 1 |
| Admission diagnosis | 19 (47.5) | 4 (100.0) | 23 (52.3) | 0.111 |
| Condition at discharge including feeding status | 19 (47.5) | 2 (50.0) | 21 (47.7) | 1 |
| Lab tests done and their results | 15 (37.5) | 0 | 15 (34.1) | 0.282 |
| Reason for hospitalization | 12 (30.0) | 0 | 12 (27.3) | 0.562 |
| Length of hospital stay | 6 (15.0) | 4 (100.0) | 10 (22.7) | <0.05* |
| Instructions for monitoring/ reasons to return to facility | 1 (2.5) | 0 | 1 (2.3) | 1 |
| Immunization status | 1 (2.5) | 0 | 1 (2.3) | 1 |
| Median (IQR) 14 item discharge form score | 8 (7-9) | 10 (9-10) | 8 (8-9) | 0.040* |

Missing data: [a]n=64, [b]n=179, [c]n=112, [d]n=63, [e]n=47, [f]n=40

Abbreviations: IQR = interquartile range

*Statistically significant p-value <0.05

**Fig 3. Observed versus reported discharge practices, N = 180 case observations at 36 facilities.**

## Discussion

Our study assessed discharge related resources and care practices across hospitals and health centers, identifying key areas where improvement could help to address the high post-discharge mortality rate among children. Resource availability assessments revealed that hospitals were generally better equipped than health centres across all domains, except for Forms & Job Aids. There were no significant differences in resource domain scores between ownership types. All facilities reported notable gaps in availability of equipment required for optimal discharge care. This was evident in the inconsistent performance and documentation of physical measurements at both hospitals and health centres, even when resources were available. The use of simple tools to identify children at risk of poor post-discharge outcomes is increasingly being developed for use in similar settings [24–27]. Many features used in these models were routinely captured though the lack of pulse oximeters, EMRs, and other technological resources highlights important barriers to scaling embedded processes which leverage such innovations.

  Our study revealed a lack of job aids across all facilities. Job aids are particularly valuable in settings with frequent understaffing and limited health worker expertise and training in discharge planning. For example, in this assessment only a single facility had a discharge policy, or a follow-up referral form, and none had counselling guides to help facilitate the discharge process. The absence of material such as these may contribute towards lower satisfaction with the discharge process among health workers [28,29] While resources undoubtedly represent a critical component necessary for the provision of quality care, many other factors, including leadership, good governance, and effective and continuous training also heavily influence care quality [28]. We found that despite fewer resources, health centres scored significantly higher in discharge counselling and discharge clinical assessments than hospitals.

Observing discharge care practices provides valuable insight into facility performance beyond resource availability. We identified discrepancies between observed and reported discharge practices across all settings, with both hospital and health centre staff inconsistently adhering to their facility's reported discharge procedures, such as providing discharge counselling, distributing take-home materials, and scheduling follow-up referrals. Interestingly, some health workers independently scheduled follow-up referrals, even though it was not reported as a discharge practice at their facility. This behaviour may reflect Uganda's Ministry of Health policy of staff rotation and transfers every 3 years, which expose health workers to different care procedures and leadership styles that promote continuous learning [30]. These findings underscore the need for standardized discharge policies and highlight the importance of job aids and discharge training to ensure all health workers are equipped with the knowledge and skills for effective discharge care, regardless of rotation or transfers.

Satisfaction among both care providers and recipients is a critical component of effective care delivery [31]; however, facility assessments often overlook health worker and caregiver perspectives [32,33]. Our findings revealed that both caregivers and health workers reported higher satisfaction with discharge care at health centres compared to hospitals. This may suggest that discharge counselling, which health workers provided more frequently at health centres, may boost overall satisfaction. Differences in staffing as well as cultural differences in care practices, patient-provider interactions, and community expectations across service delivery levels [34] may account for these findings and warrants further exploration. Caregiver perspectives matter as those who feel respected and valued are more likely to adhere to follow-up care [31]. Moreover, caregiver satisfaction is associated with interventions that enhance care coordination and family engagement, such as scheduling follow-up care, providing written care plans, and offering parental education [35].

Employee satisfaction provides valuable insight into working conditions and an organization's overall preparedness. In our study, only half of health workers reported being "always" or "often" satisfied with their facility's discharge process, with higher satisfaction at health centres. Factors such as a closer interpersonal relationship between health workers, family members, and the community, may account for greater satisfaction among health centre staff by fostering a collaborative work environment and a sense of trust [34,36]. Improving discharge care through standardized discharge policies, comprehensive training, and community referrals, as suggested by health workers in other studies, may further enhance satisfaction [16].

## Limitations

This study is subject to several limitations. First, our facility assessment tool lacks extensive validation and is not widely used or promoted like other established tools such as the WHO's SARA [37], USAID's Service Provision Assessment (SPA) [38], or the WHO's Harmonized Health Facility Assessment (HHFA) [39]. However, these tools have limited focus on the discharge and post-discharge experience, despite the increasingly recognized role that robust health systems play in reducing post-discharge mortality. Second, our district-level sampling frame may limit the generalizability of our findings to the whole of Uganda due to overrepresentation of two specific districts. However, it is possible that differences between districts exceed our observations of the differences within districts, though the wide variation within districts suggests that for this to be true, the differences would need to be quite significant. Given this, we do not believe that our sampling significantly biased our results as they pertain to country-level inferences. Third, the observation component was limited by the fact that some clinical assessments (e.g., pallor, respiratory distress) may have been performed by the clinician but not easily observed by study staff, or perhaps not been indicated given the clinical situation. A think-aloud approach [40] may mitigate this kind of limitation but may likewise result in further deviation from typical care that would be provided absent of an observer. Moreover, the presence of study staff may have altered health worker behaviours during the observation period (Hawthrone effect) [40,41]. Similarly, while we recorded whether clinical assessments were conducted at discharge, we did not capture the results of these assessments or specify which assessments were performed. Lastly, this study was conducted during the COVID-19 pandemic which may have impacted resource availability due to supply

chain issues as well as satisfaction with discharge care due to staffing shortages, potentially introducing bias into our results. Despite these limitations, the findings offer valuable insights that can substantially contribute to the enhancement of discharge policies, with implications extending beyond the immediate scope of suspected sepsis care.

### Strengths

This study has several key strengths. We utilized a nationally representative sample of health facilities, providing a comprehensive overview of discharge care across diverse settings in Uganda. Notably, the study incorporated a thorough assessment of resource availability, procedures, and care practices specifically related to hospital discharge, an area that is often overlooked in existing facility assessments. Additionally, the inclusion of both health worker and caregiver satisfaction provides valuable insights into the human factors influencing care, which are frequently neglected in facility assessments. These strengths contribute to a more holistic understanding of discharge processes and their impact on care quality.

### Conclusion

The findings from this facility scan have pinpointed key areas for quality improvement in both resource availability and effective provision of discharge care. Despite having fewer resources, health centres demonstrated better discharge care counseling practices and higher satisfaction metrics than hospitals, highlighting the need to investigate factors contributing to these successful practices and exploring their applicability in hospitals. Targeted interventions, such as standardized discharge policies that encompass objective risk assessments, discharge counselling, take-home materials, and referrals, as well as improved health worker training and job aid availability, are crucial to address the challenges identified. These insights can serve as a roadmap for shaping a national-level pediatric discharge policy in Uganda which may ultimately improve post-discharge outcomes.

### Supporting information

**S1 Text. Details of sampling and facility scan and detailed tables and figures of facility scan results.**
(DOCX)

**S1 Checklist. Inclusivity in global research.**
(DOCX)

### Acknowledgments

The authors express their gratitude to Okeny Louis and Tusingwire Fredson for their research assistance and to Peter Lewis for providing technical support.

### Author contributions

**Conceptualization:** Jessica Trawin, Clare Komugisha, Jesca Nsungwa Sabiiti, J. Mark Ansermino, Niranjan Kissoon, Nathan Kenya Mugisha, Matthew O. Wiens.

**Data curation:** Jessica Trawin.

**Formal analysis:** Jessica Trawin, Martina Knappett, Clare Komugisha, Matthew O. Wiens.

**Funding acquisition:** Nathan Kenya Mugisha, Matthew O. Wiens.

**Investigation:** Clare Komugisha, Ezrah Bamwesigye, Collins Agaba.

**Methodology:** Jessica Trawin, Clare Komugisha, J. Mark Ansermino, Niranjan Kissoon, Nathan Kenya Mugisha, Matthew O. Wiens.

**Project administration:** Jessica Trawin, Clare Komugisha.

**Resources:** Jessica Trawin, Nathan Kenya Mugisha, Matthew O. Wiens.

**Supervision:** Nathan Kenya Mugisha, Matthew O. Wiens.

**Validation:** Jessica Trawin, Martina Knappett, Clare Komugisha.

**Visualization:** Jessica Trawin.

**Writing – original draft:** Jessica Trawin, Martina Knappett, Clare Komugisha, Niranjan Kissoon, Matthew O. Wiens.

**Writing – review & editing:** Jessica Trawin, Martina Knappett, Clare Komugisha, Savio Mwaka, Ezrah Bamwesigye, Collins Agaba, David Walugembe, Jesca Nsungwa Sabiiti, J. Mark Ansermino, Niranjan Kissoon, Nathan Kenya Mugisha, Matthew O. Wiens.

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
