## [Decision Letter · Decision Letter 0]

18 Sep 2024

PGPH-D-24-01608

Discharge policies and care practices for children with suspected sepsis: A health facility scan at a nationally representative sample of hospitals and health centres in Uganda

Dear Dr. Trawin,

Thank you for submitting your manuscript to PLOS Global Public Health. After careful consideration, we feel that it has merit but does not fully meet PLOS Global Public Health’s publication criteria as it currently stands. Therefore, we invite you to submit a revised version of the manuscript that addresses the points raised during the review process.

This research represents an important and underpublished topic that sheds light onto discharge practices for children admitted with sepsis in Uganda. The manuscript needs significant edits particularly to strengthen the methodology and conclusions and to improve clarity throughout. Please address the excellent reviewer comments and we look forward to reviewing a revised version. 

We look forward to receiving your revised manuscript.

Kind regards,

Heather Haq, M.D., M.H.S.

Academic Editor

Journal Requirements:

1. Thank you for providing the requested questionnaire for global inclusivity. However, we note that the questionnaire has been submitted as an ""other"" file type. Kindly resubmit the global inclusivity questionnaire as Supporting file.

2. In the online submission form, you indicated that "Access to these data will be granted on a case-by-case basis following approval from the authors and the Data Governance Committees."

3. Uploaded as supplementary information.

Additional Editor Comments (if provided):

This research represents an important and underpublished topic that sheds light onto discharge practices in Uganda. The manuscript needs significant edits particularly to strengthen the methodology and conclusions.

Reviewers' comments:

Reviewer's Responses to Questions

**Comments to the Author**

1. Does this manuscript meet PLOS Global Public Health’s publication criteria? Is the manuscript technically sound, and do the data support the conclusions? The manuscript must describe methodologically and ethically rigorous research with conclusions that are appropriately drawn based on the data presented.

Reviewer #1: Yes

Reviewer #2: Partly

Reviewer #3: Yes

2. Has the statistical analysis been performed appropriately and rigorously?

Reviewer #1: Yes

Reviewer #2: No

Reviewer #3: I don't know

3. Have the authors made all data underlying the findings in their manuscript fully available (please refer to the Data Availability Statement at the start of the manuscript PDF file)?

Reviewer #1: Yes

Reviewer #2: Yes

Reviewer #3: Yes

4. Is the manuscript presented in an intelligible fashion and written in standard English?

Reviewer #1: Yes

Reviewer #2: No

Reviewer #3: Yes

5. Review Comments to the Author

Reviewer #1: This paper assesses the readiness of health facilities in Uganda to manage pediatric discharge care for children under 5 years or age, focusing 36 representative health facilities. The study employed five distinct surveys to assess facility resources, technological preparedness, quality and safety during severe illness/infection, patient-caregiver perceptions, and health worker perceptions of care. The findings reveal discrepancies in discharge counseling practices between health centers and hospitals. Specifically, health centers had more robust discharge counseling and higher caregiver satisfaction despite lower infrastructure domain scores, whereas hospitals provided more post-discharge risk assessments and scheduled follow up more frequently. These results identify critical areas for quality improvement and can offer valuable guidance for shaping national level pediatric discharge policy in Uganda. The study has an impressive scope and is well-designed, thorough, and well-written, making it a strong candidate for publication.

Recommendations:

Intro:

1. Clarify that the study specifically targets pediatric discharge care for children under 5 years old in sections such as the abstract and introduction to make it clear for the readers.

2. Page 4, line 70: explicitly state how the study aligns with or utilizes WHO guidelines for effective hospital discharge, as this connection is currently unclear.

Methods:

3. Define the primary outcome more clearly. The term “discharge readiness” is unclear without a defined metric or validated scales. Consider focusing on outcomes such as resource availability and the observation of discharge care practices. Ensure these primary outcomes are referenced in the abstract and conclusion/discussion for clarity.

4. Please include the modified questions from the Sepsis CoLab Environmental Scans in the supplemental files, as these modifications pertain to discharge care. Currently, they are under moderated access.

5. If an assessment of pediatric vital signs at discharge was conducted (ie obtained vs not obtained, normal vs abnormal), please include this information. If not, consider discussing the absence of this data in the discussion section.

Results:

6. Infrastructure domain scores are first mentioned on Page 11, line 208. Please provide a detailed explanation in the methods section regarding how these scores were derived and what they encompass.

7. Page 10 ,Lines 195-196: clarify the discrepancy in patient numbers (n=180 discharged patients vs. n = 145 in the following line). Consider revising the total N to 145 if appropriate.

8. Address if any enrolled facilities were omitted from the dataset for observation of discharge care practices (as n = 145) and discuss how this might affect the study’s findings.

9. Page 10, Lines 197 (or Figure 1): indicate the nature of the diagnoses for the remaining 37.9% of patients not treated for malaria or pneumonia, as this represents the second-largest category.

Other comments:

A thorough copy-editing is recommended to correct minor spelling and grammatical errors. For instance, line 52 should use “sub-Saharan” rather than “Saharan”, line 204: “enrolment” is misspelled, line 271-273 “whereas…compared to hospital health workers.” should be a complete sentence, Table 2 should list “n” as 65 under “discharge counseling topics, hospitals”. Additionally, consider improving the quality of the figures, which appear slightly grainy.

Overall, this manuscript provides significant insights into pediatric discharge readiness in low and middle income countries and has the potential to make a substantial contribution to the literature.

Reviewer #2: Major comments and revisions:

- It’s overall unclear to me what the big take-home messages are, from this analysis, that are actionable either at a policy level or in the next phase of research into this problem.

- Would recommend having someone do a general edit for writing clarity and appropriate use of abbreviations (most notably “n”). The paper needs a close grammar check; some sentences are awkwardly worded and subject-verb agreement is sometimes off. Also would change passive voice to active voice throughout; the passive voice makes it much more difficult to follow your methods and results.

- Clarify criteria for any purposive sampling, and qualitative methods generally.

- It’s unclear to me how the authors have determined that this is a nationally representative sample; this assertion needs further explanation, as they have excluded one segment of the health facilities and have not represented all regions of the country.

- Decrease references to supplementary materials; consider separating into multiple manuscripts if there are so many important results so as to be difficult to pare down to a reasonable number of tables and figures for inclusion in the text.

- Avoid reporting trends in data that were not significant, especially with this much data to parse.

- Qualitative methods and themes are inadequately described. What were your qualitative questions?

- Please address whether there are any specific conclusions about particular policies that might impact post-discharge mortality (e.g., follow-up appointments not being routine), that aren’t currently widely in place? Are there any SPECIFIC interventions that emerge from your data, that Uganda could trial?

Minor comments and revisions:

Abstract:

What is a “facility scan?” Did you use a validated instrument?

Were private for-profit facilities assessed?

Introduction:

Line 59 – typo: “require,” not “requires”

You note that little work has been done in LMICs on discharge practices, but one of the only papers you cite that supports the assertation that discharge his a “complex and error-prone process” is actually from Uganda. Is there other literature on the discharge process from HICs that you could cite, or do we just not know a lot about what makes a safe discharge in general?

Line 58 typo – “provide” not “provides”

Methods:

Why were private for-profit health facilities excluded? How can you show that the decision to exclude them did not impact the socioeconomic and cultural representativeness of your sample?

How did you select the health facilities? You said purposively, but using what criteria?

How were the 2 districts selected?

It’s unclear from your methods how you’ve reached the conclusion that your data is nationally representative, given that it excludes representation of lower-level health facilities in large parts of the country.

You say that caregivers provided written informed consent - did you exclude limited-literacy caregivers, or did you provide them support in order to participate?

Please further describe any criteria you used to purposively select your health worker participants.

Line 150, 153, 156: closed-ended, not close-ended

Not sure that “n” is being used appropriately to refer to the number of items in the score -was each item worth one point? I believe that “n” should be the number of records assessed, and the scoring system should be described separately. Is “points” the unit on the median and IQR? There is also variability in how “n” is being used that should be cleared up.

No dash between the number and “items.”

Qualitative methods are inadequately described (how were codes determined? How many coders? Who were they? What was the reconciliation process?)

The sample size may not be big enough, but it would be helpful to separate out each level of care into private vs. public, to see if different patterns emerge between types of facilities when looking at different levels of the health system.

Results:

I generally found the results difficult to follow due to unclear phrasing, excessive passive voice, and difficulty differentiating among the different tools used and what their various purposes were. There is also widely variable and inappropriate use of “n” to describe various numbers. Generally this section needs to be edited to the most important findings; the authors might consider including others in a second manuscript.

What does “routinely discharged” mean? (line 196)

Why is the only demographic information reported for the caregivers the distance they travelled to the hospital/health centre? Were other characteristics collected? If so, these would help us understand the socioeconomic and other demographics of the population studies, which is likely to be important, especially when determining whether this is truly a nationally representative sample.

Pretty impressive that 80% of facilities provided meds for free. Are meds supposed to be free by law or policy in the public system in Uganda?

Line 217: I can’t follow what this line means: “Facilities having referral clinics for 218 post-discharge follow-up was more commonly reported among hospitals (56.5%, n=13) than 219 health centres (7.7%, n=1) (p=0.011) and PNFP facilities (50%, n=4) compared to public 220 (35.7%, n=10) (p=0.683).”

Results should be formatted: (median = 3; IQR = 2 – 4) or simply (m=3, IQR=2-4)

Line 34: typo “owrnership”

Line 241: “Although socio-economic history taking was not commonly performed and documented among all cases, domain scores (n=7 items) were significantly higher (p<0.001) among health centres (median 4: IQR: 2 – 4) compared to hospitals (median 1; IQR: 0 – 2) (S2 Fig, S3 Table).,” can be simplified to something like: “Socio-economic history was infrequently documented overall. However, in health centres the median socio-economic history-taking score was 4 (IQR=2-4), higher than the median score at hospitals (median=1, IQR=0-2) (S2 Fig, S3 Table).”

Related note: how do you know that indicators were infrequently performed? Does the score reflect both whether the health care worker took the history and also whether it was documented? What’s the significance of including both of those measures?

Line 332: “insignificant” should be “not significant.” Also, if it is not statistically significant, it is inaccurate to say that it was higher. You could say that it “trended toward” being higher, though that’s also usually not advised, because the statistical analysis does not support it.

It would be helpful to describe qualitative themes more systematically. You may also need a frequency table for codes or themes, or a weighting schema in your analysis, since you report that several things are “commonly reported” but don’t explain exactly what you mean by that, or which were the most prominent themes in your analysis.

Line 340: “Health workers were asked to rate staff availability for several discharge scenarios at their facility.” This belongs in the methods, along with richer description of what the various scales are assessing.

Discussion:

Line 352: Incomplete sentence: “This is not surprising and in view of the high mortality rates post-discharge.”

Lines 384-385: It’s unclear to me how health workers doing something that isn’t part of the formal discharge process supports the importance of job aids and education in discharge planning.

Line 393: What “cultural differences?”

Separate out limitations in a new sub-section.

Add “strengths” sub-section.

This is a big assumption: “However, it is unlikely that differences between districts would vastly exceed differences within districts, and thus, we do not believe that our sampling significantly biased our results as they pertain to country-level inferences.” Is there any data from other studies to back this up and support the generalizability of your findings?

Discussion of the limitations of the observation component is difficult to contextualize, as the observations aren’t really described, so it’s unclear how big an impact this limitation might have had.

What is a “think-aloud approach?”

If possible, it would help to better characterize COVID-related resource and staffing shortages in these clinics, to help the reader understand the context and limitations of the study better.

Reviewer #3: Overall, this is a thorough look at discharge practices in selected health centres and hospitals in Uganda. The authors clearly worked hard to gather the data presented here. However, there is a lot presented. This manuscript could benefit from some editing to make everything more concise, or by removing some of the findings to write them up separately. Specific comments by section below:

Abstract

- Results: can you give more info on “infrastructure domain scores” or use the same language as in the methods?

-

Intro

- In line 52, I think it should be “sub-Saharan Africa” given the abbreviation that follows

- The WHO info (lines 67-70) is good information but maybe not placed well; consider moving it to the 2nd paragraph, perhaps around line 61

Methods

- You give a lot of background in the types of health centres/hospitals in Uganda, then simply combine them into health centres and hospitals. You could simplify this paragraph somewhat since the division of type of health centre or hospital isn’t used in analysis

- I’m not sure I understand the district vs national sampling used. Are there more lower level health centres in those 2 districts and that is why more were chosen from 2 districts? Is the population different in those 2 districts vs the other 20? A bit more explanation would be helpful

- Under participant sampling and recruitment: is there a reason enrollment was limited to those admitted for suspected sepsis?

o Did the ability to provide written informed consent limit enrollement? What language was the informed consent document written in?

- The number of different scores used (lines 161 onward) gets somewhat confusing, if there is any way to simplify this

- Were the observational surveys done on the same children that caregivers completed surveys about?

Results

- Line 196, What do you mean by “routinely discharged”? What happened to the other 2%?

- Line 213-214, you mention the 29 provided for free and 8 provided meds at cost; that doesn’t add up to 36. Was there overlap where some meds were free and others at cost?

- Line 217, the % and N for the public facilities doesn’t make sense; if it’s 96% of them with stock outs then the n should be higher than 1.

- For the admission section, if there are 180 observations, is it 180 unique patients or unique observations? Was each patient observed at admission and discharge to comprise 1 observation?

- Line 299, you report that 19.9% of health centre cases were not provided with take-home materials, but in table 2 you report that only 4% received take home materials, so this doesn’t add up; the numbers for hospitals also don’t add up

- Overall results section is very lengthy and could benefit from trimming down some of this information or perhaps even dividing it into 2 manuscripts (the survey and observation I one, the caregiver/health care worker satisfaction in another). The high number of tables and figures make it difficult to follow the text sometimes

Discussion

- Again, all the different names for the different surveys/scans gets confusing (line 353)

- The resuls section spends a lot more time looking at the care practices/satisfaction than the resource availability, but the discussion spends more time looking at resources and ways to scale up resources (makes sense that improving resources could improve the rest of it, but the comment in line 375 seems to indicate that resources are less important)

- Can you comment on the disconnect between reasonably high satisfaction levels but low overall discharge preparation/guidance?

- Appropriate discussion of limitations

Conclusion

- Your results seem to argue more that health centres provide better counselling while hospitals do better with materials, risk assessment, and follow up referrals which doesn’t necessarily mean that health centres do better with discharge care overall (although satisfaction levels imply they may)

Tables

- Overall, there are way too many tables and figures in this manuscript. Is there any way to combine or remove some of them? The supplemental figures are referenced pretty frequently without a lot of description, and it can sometimes be difficult to find supplemental figures after publication

o You could combine S2, S3, S4, S5, S6 much like S1 or remove some of those and combine the rest

S3 table, S4 table also appear to be in another form elsewhere and therefore repetitive

o S7, S8 are included in Table 2 and not necessary

o S1 Table is less informative than S2 Table

- Table 1; if you simplify some the description this could be a supplemental table if it’s used at all, freeing up a space for another results table

- Table 2, under discharge documents provided, n for discharge form for hospitals is 41, but only 40 in the next section of the table

6. PLOS authors have the option to publish the peer review history of their article (what does this mean?). If published, this will include your full peer review and any attached files.

**Do you want your identity to be public for this peer review?** For information about this choice, including consent withdrawal, please see our Privacy Policy.

Reviewer #1: No

Reviewer #2: No

Reviewer #3: No

---

## [Decision Letter · Decision Letter 1]

7 Apr 2025

PGPH-D-24-01608R1

Discharge policies and care practices for children with suspected sepsis: A health facility scan at a nationally representative sample of hospitals and health centres in Uganda

Dear Dr. Wiens,

Thank you for submitting your manuscript to PLOS Global Public Health. After careful consideration, we feel that it has merit but does not fully meet PLOS Global Public Health’s publication criteria as it currently stands. Therefore, we invite you to submit a revised version of the manuscript that addresses the points raised during the review process.

Your revision represents substantial improvements to the manuscript that addressed many of the reviewers' initial critiques; however, it still requires additional revisions before it can be accepted for publication. Please address the attached suggestions from the two reviewers.

We look forward to receiving your revised manuscript.

Kind regards,

Heather Haq, M.D., M.H.S.

Academic Editor

Journal Requirements:

Additional Editor Comments (if provided):

Reviewers' comments:

Reviewer's Responses to Questions

**Comments to the Author**

1. If the authors have adequately addressed your comments raised in a previous round of review and you feel that this manuscript is now acceptable for publication, you may indicate that here to bypass the “Comments to the Author” section, enter your conflict of interest statement in the “Confidential to Editor” section, and submit your "Accept" recommendation.

Reviewer #1: (No Response)

Reviewer #2: (No Response)

2. Does this manuscript meet PLOS Global Public Health’s publication criteria? Is the manuscript technically sound, and do the data support the conclusions? The manuscript must describe methodologically and ethically rigorous research with conclusions that are appropriately drawn based on the data presented.

Reviewer #1: Partly

Reviewer #2: Yes

3. Has the statistical analysis been performed appropriately and rigorously?

Reviewer #1: Yes

Reviewer #2: Yes

4. Have the authors made all data underlying the findings in their manuscript fully available (please refer to the Data Availability Statement at the start of the manuscript PDF file)?

Reviewer #1: Yes

Reviewer #2: Yes

5. Is the manuscript presented in an intelligible fashion and written in standard English?

Reviewer #1: Yes

Reviewer #2: Yes

6. Review Comments to the Author

Reviewer #1: This manuscript assesses the readiness of health facilities in Uganda to manage pediatric discharge care for children under 5 years or age, focusing on 36 representative health facilities. The authors have made significant revisions in response to reviewer comments, which have strengthened the paper.

Due to its scope and design, as well as its potential to impact practices in LMICs, it continues to be a strong candidate for publication.

The manuscript would benefit from improved organization and consistency in reporting the outcomes, which would make it easier for readers to follow significant findings.

Recommendations:

Abstract:

- Ensure consistency between the abstract methods, results, and conclusion and the manuscript in terms of primary and secondary outcomes, and findings based on those outcomes.

Methods:

- Line 84: Consider including total N for hospitals in Uganda, as the total number of admitting facilities is provided.

- Consider clarifying the difference in services between health centres and hospitals, as both types of facilities admit patients (unlike in the United States, where typically only hospitals admit). This distinction could be in the methods or introduction.

- Was immunization and record keeping considered, as it is noted as part of the four key WHO guidelines for effective hospital discharge? If not, this could be addressed in the discussion or limitations section.

- Please provide further clarification on how the primary outcomes were determined. Specifically, how was the resource availability score calculated? How was observation of the four key discharge care practices quantified? How are resource availability and observation scores combined to determine overall facility readiness for discharge? It would also be helpful to explain the rationale for why these indicators were chosen to represent the facility readiness for discharge score. It would also be helpful to clarify what defines a good outcome for facility readiness for discharge.

- There is some inconsistency in the description of the primary outcome. In the study outcomes section (Lines 146-147), facility readiness for discharge is defined as both resource availability and observation of discharge practices. However, in the statistical analysis section (Lines 170-175), it states that facility readiness scores are calculated based solely on resource availability. Please clarify which of these is accurate

- The methodology for calculating care practice domain scores from the seven domains needs further clarification, as well as how these relate to the four key discharge care practices listed in the primary outcomes. It would be helpful to explicitly outline this for the reader.

- Clarify how the secondary outcome of observed versus reported discharge care practices is calculated and how it differs from the primary outcome of “observation of four key discharge care practices.”

Results:

- Consider presenting results of the primary outcome, facility readiness for discharge, before breaking it down into its components.

- Consider presenting the results of overall resource availability scores before breaking them down into individual domains (e.g., infrastructure, technology, forms & job aids, equipment). This would help with the interpretability of the results.

- Similarly, present the overall care practice scores before detailing the breakdown of specific practices.

- Consider restructuring Table 2 to align more clearly with the primary outcomes. Currently, it takes time for the reader to align the subheadings in the table with either the four key discharge care practices or the seven domains presented in the statistical analysis.

Discussion:

- Consider summarizing all pertinent findings regarding the primary outcomes in the first paragraph before discussing the individual components. This would provide clarity for the reader and highlight key findings early in the discussion.

Other comments:

1. In Table 2, consider highlighting significant p-values (for example, by putting an asterisk or bolding them)

2. Consider copy-editing for minor spelling errors. For example, Line 279 “In contract”, should be “In contrast.”

Overall, this manuscript provides valuable insights into pediatric discharge readiness in low and middle income countries and has the potential to make a substantial contribution to the literature.

Reviewer #2: I appreciate that the authors have spent significant time revising the manuscript, and the overall study is extremely worthy and very interesting, shedding light on an important and oft-overlooked element of care. They have addressed many of my previous comments and overall the paper is much improved and easier to read. In my opinion, though, it is still not quite ready for publication, primarily because the many scores, percentages and referenced tools still make it difficult to follow, and the references to supplementary tables are still too frequent. I needed to have two separate pdfs of the manuscript open in order to read it, because following all the details required so much scrolling and re-reading, especially when reviewing the results section.

I would suggest that the writing team use their primary tables and figures (which should present the most important findings) to guide the writing, and either provide all other data in a set of comprehensive supplementary tables that are NOT necessary to reference to understand the main results, or separate into several manuscripts. The presented findings should also correlate to the specific study objectives, with very sparing presentation of interesting findings from individual questionnaires or response fields.

Further comments by section:

Abstract:

Study objectives are not presented as presented in the writeup; please include these, rather than describing the facility scans, as the facility scans have been broken down (as I understand it) in a different way to answer the study questions.

Why is the p value for post discharge counselling presented as <0.021? Is this an error?

Introduction:

Consider introducing the Environmental Scan and its previous uses in the introduction, rather than discussing the WHO guidelines, which are never mentioned again, and it's unclear how exactly they are related to the scan used in the methods.

Methods:

Sampling description is much improved! Thank you.

The authors responded in the comments to reviewers as to why for-profit hospitals were excluded but still do not detail the reason for this exclusion in your methods, which I would argue does need to be addressed, just with a one-sentence explanation.

Would move description of your data collection tools above your participant recruitment, because you currently reference the questionnaires before you tell us what questionnaires you use, and it reads as out-of-order.

Thank you for more fully describing the Sepsis CoLab environmental scan.

This reads as though informal consultation was the only method used to write discharge modules. Understanding that this may be the case and may be related to resource or time constraints, if there were more formal methods (focus groups, delphi method, etc) used to write these modules, they should be included here (lines 136-143) or referenced if published elsewhere.

The elements of the surveys that are used to generate each outcome are not a statistical method, and so in my opinion should be described in your Study Outcomes section, not your statistical analysis section. This would also help clarify the relationship between your outcome measures and your study instrument. This clarification might also be achieved by just breaking up your paragraphs more, and starting each with a topic sentence that reads "xxx outcome measure was calculated using xxx scans." But also, methodologically is it actually important which scan each of the measures comes from? Would it be possible to just say "We compiled scores using items from the 5-survey Facility Scan..." without going into so much detail about the scans themselves, especially in the statistical methods? Does it matter which survey the measure comes from, as--as I understand it--you recombined them all anyway? Think about it as treating the Environmental Scan as one tool rather than a set of 5 tools. You can include the details in a supplement because they are not essential for understanding your outcomes, and so if the reader can't or doesn't want to access them, it's ok.

I do not understand this line: Missing data were removed from the analysis and reported throughout the manuscript as numerators over denominators to clarify the variables affected by missing data. Are the missing data in discharge observations, for example, the difference between 180 and 145? Where are the numerators and denominators?

Thank you for further describing your informed consent practices.

Results:

If continuing to reference the separate surveys, would it be possible to include a table that summarizes which pieces of which survey were categorized into which domain for analysis, along with the scores in each? You present a lot of this data, but it’s hard to follow because it’s very complex and there are so many categories.

Presentation of proportions and use of N vs n continues to be unclear, and nearly uninterpretable without very close reading or referencing your figures. Examples and suggestions to further simplify this for the reader:

1. When presenting a proportion, I don't think there's any need to present it as "n=," because you're not presenting the number, but the proportion. For example, "most caregivers (n=57/145; 39%) can more clearly be presented as "39% (57/145) of caregivers interviewed."

2. For three categories on your flow chart, N is variably the total population of 180 discharge observations or interviews, vs the number of patients analysed UNDER each variable (sex, discharge status, discharge diagnosis, travel time, etc), with n representing all subgroups in all analyses, and the capitalization is not used uniformly in the manuscript (N vs n, example line 211 where n does not match the N used in the table to reference the same population). Interpreting which N/n you’re talking about is very difficult without referencing the chart. On line 210, n references a percentage! May consider using subscripts (Nrespondents vs Ntotal, for example, or defining different variable letters for better clarity.

Why are all the Ns for the discharge characteristics 145 when you have 180 observations? Is this just a coincidence, or were that actually only 145 analyzable surveys? So is 145 your actual analyzed population, not the 180 observations? It becomes clear later that the 180 observations yielded other important data, but it’s not clear here where you present your demographic results that that is why the 180 number is important.

As mentioned in a prior comment, I recommend avoiding “trended toward” when discussing results that were not statistically significant - eg line 239. Your IQRs are the same, even if the medians are different, and your p-value is nonsignificant, so your data is not supportive of a difference.

Where are the admission care practice score integrated into your analysis of discharge readiness? It does not seem to be integrated into your stated primary or secondary outcomes, and all of the results you mention are in supplementary tables and figures - are they really critical to present here or should they be analyzed and written up separately? This is very interesting information worthy of reporting in the literature, perhaps in the context of looking at correlation between admission practices and discharge practices, but is it relevant to your outcome measures here?

Figure 3 is excellent and very clear! This is the place where I best understand your findings.

Discussion:

I think the discussion is quite strong. Of note, I don't think I see the admission care practices reflected anywhere in it, which to me feels like another argument to describe those in a separate venue, or else incorporate them more thoroughly into your outcomes.

Thank you for addressing the national representativeness of the sample in your limitations; this adds strength and transparency to the paper in my opinion.

Some copyedits:

Standardize under-5 or “under 5” mortality. “Under-5” is standard when the phrase is used as an adjective; when it is phrased as “children under five,” there is no hyphen.

Spell out all non-decimal numbers under 10. (see line 90)

“For-profit” and “not-for-profit” should be hyphenated.

Define the acronym PFNFP on first use of the phrase on line 86.

Period missing line 87.

Health care should be two words in AMA style.

In-patient should be inpatient - line 100 and throughout.

Figures and Tables:

These are very nice and where your main argument is made most effectively, even with the difficulties noted about the flow chart.

7. PLOS authors have the option to publish the peer review history of their article (what does this mean?). If published, this will include your full peer review and any attached files.

**Do you want your identity to be public for this peer review?** For information about this choice, including consent withdrawal, please see our Privacy Policy.

Reviewer #1: No

Reviewer #2: No

---

## [Decision Letter · Decision Letter 2]

17 Jul 2025

PGPH-D-24-01608R2

Discharge policies and care practices for children with suspected sepsis: A health facility scan at a nationally representative sample of hospitals and health centres in Uganda

Dear Dr. Wiens,

Thank you for submitting your manuscript to PLOS Global Public Health. After careful consideration, we feel that it has merit but does not fully meet PLOS Global Public Health’s publication criteria as it currently stands. Therefore, we invite you to submit a revised version of the manuscript that addresses the points raised during the review process.

We look forward to receiving your revised manuscript.

Kind regards,

Miquel Vall-llosera Camps

Staff Editor

Journal Requirements:

Reviewers' comments:

Reviewer's Responses to Questions

**Comments to the Author**

1. If the authors have adequately addressed your comments raised in a previous round of review and you feel that this manuscript is now acceptable for publication, you may indicate that here to bypass the “Comments to the Author” section, enter your conflict of interest statement in the “Confidential to Editor” section, and submit your "Accept" recommendation.

Reviewer #1: (No Response)

2. Does this manuscript meet PLOS Global Public Health’s publication criteria? Is the manuscript technically sound, and do the data support the conclusions? The manuscript must describe methodologically and ethically rigorous research with conclusions that are appropriately drawn based on the data presented.

Reviewer #1: Yes

3. Has the statistical analysis been performed appropriately and rigorously?

Reviewer #1: Yes

4. Have the authors made all data underlying the findings in their manuscript fully available (please refer to the Data Availability Statement at the start of the manuscript PDF file)?

Reviewer #1: Yes

5. Is the manuscript presented in an intelligible fashion and written in standard English?

Reviewer #1: Yes

6. Review Comments to the Author

Reviewer #1: This manuscript assesses the readiness of health facilities in Uganda to manage pediatric discharge care for children under 5 years or age, focusing on 36 representative health facilities. The authors have made significant revisions in response to reviewer comments, especially clarifying the primary and secondary outcomes, which have strengthened the manuscript.

Due to its scope and design, as well as its potential to impact practices in LMICs, it continues to be a strong candidate for publication. A few areas would benefit from additional clarification and consistency.

Recommendations:

Methods:

- The methodology for deriving the second primary outcome needs further clarification. Specifically, it is unclear how the five key discharge practices were calculated as binary variables when they are derived from seven domains. Please clarify whether a certain number of items from each domain needed to be present for a practice to be considered “present”, or whether the five key practices were predefined as binary outcomes with Scan 3 subsequently assessing greater detail within three of them, or another method. An explicit description would help improve transparency for the reader.

- What components constitute a post-discharge risk assessment? A concise definition can help readers interpret this outcome.

- Please clarify in the methods how these two secondary outcomes were assessed: (1) discharge-relevant admission practices, and (2) caregiver and health worker satisfaction. Were Scans 4 and 5 used for caregiver and health worker satisfaction, respectively? If so, please state this.

- The discharge-relevant admission practice outcome is not clearly described in the Methods section and is not integrated into the Discussion. Additionally, based on Table S2 and Figures S2a-e, some components of this outcome (for example, medical history items) are unrelated to discharge care, although it is true that discharge preparation begins at admission. The authors can consider removing this secondary outcome and associated supplementary tables/figures to streamline the manuscript and focus on the many other findings already described, or clearly incorporate it in the Methods and Discussion section.

- Please ensure that the categories of the second primary outcome and secondary outcomes are listed in the same order across the abstract, methods, results, discussion, and tables for reader comprehension.

Results:

- Please indicate how many hospitals and health centres are represented in each of the PNFP and public categories. This could be added after the first sentence in the Results (Line 206) and listed in Table 1 under the respective headings.

- Lines 224-241, lines 259-277, (and lines 291-310 if keeping in the manuscript): The number of statistical comparisons make interpretation difficult. Consider highlighting only the most relevant findings in the narrative and grouping results by domain in each of the outcomes (for example, for resource availability grouping by infrastructure, equipment, etc). If possible and where relevant, consolidate comparisons between hospitals and health centres and between PNFP and public facilities to improve readability.

- Line 236: The phrase “though most facilities had key equipment available to assess post-discharge vulnerability” is vague and potentially subjective. Consider removing or rephrasing, as definitions of “key” can vary.

- Line 272-273: Please include the numbers or p-values to support the statement that discharge forms were more complete at health centres.

- Line 278: Consider beginning this section with a new subheading, as it introduces a secondary outcome of reported vs observed discharge practices.

- Line 279: Please clarify why the fifth key discharge practice was not included in the reported vs observed comparison.

Discussion:

- First paragraph: Consider revising the opening paragraph of the Discussion to summarize the most important findings across all of the primary and secondary outcomes in order to frame the remainder of the discussion.

- Line 347-348: Please revise this sentence to accurately reflect the findings in Table 2. Based on the data, health centres scored significantly higher in discharge counseling and discharge day clinical assessments, while hospitals scored significantly higher in the remaining three discharge care practice categories. The phrase “similar findings were also noted” may also need to be revised or removed, as the findings differ across categories.

Other:

A thorough copy-editing is recommended to address minor spelling and grammatical errors. An few examples (non-exhaustive) are noted below.

- Figure 1 title: Facility enrolment-> facility enrollment

- Table 1: In the caption, remove “of resource availability.” The p-value for “facility has referral clinics that follow-up children post-discharge” should be marked with an asterisk for significance.

- Line 215: “within a 1 hour” -> “within 1 hour”

Overall, this manuscript provides important insights into pediatric discharge readiness in low and middle income countries and has the potential to make a valuable contribution to the literature.

7. PLOS authors have the option to publish the peer review history of their article (what does this mean?). If published, this will include your full peer review and any attached files.

**Do you want your identity to be public for this peer review?** For information about this choice, including consent withdrawal, please see our Privacy Policy.

Reviewer #1: No

---

## [Decision Letter · Decision Letter 3]

7 Aug 2025

Discharge policies and care practices for children with suspected sepsis: A health facility scan at a nationally representative sample of hospitals and health centres in Uganda

PGPH-D-24-01608R3

Dear Dr. Wiens,

We are pleased to inform you that your manuscript 'Discharge policies and care practices for children with suspected sepsis: A health facility scan at a nationally representative sample of hospitals and health centres in Uganda' has been provisionally accepted for publication in PLOS Global Public Health.

Best regards,

Julia Robinson

Executive Editor

Reviewer Comments (if any, and for reference):

Reviewer's Responses to Questions

**Comments to the Author**

1. If the authors have adequately addressed your comments raised in a previous round of review and you feel that this manuscript is now acceptable for publication, you may indicate that here to bypass the “Comments to the Author” section, enter your conflict of interest statement in the “Confidential to Editor” section, and submit your "Accept" recommendation.

Reviewer #1: All comments have been addressed

2. Does this manuscript meet PLOS Global Public Health’s publication criteria? Is the manuscript technically sound, and do the data support the conclusions? The manuscript must describe methodologically and ethically rigorous research with conclusions that are appropriately drawn based on the data presented.

Reviewer #1: Yes

3. Has the statistical analysis been performed appropriately and rigorously?

Reviewer #1: Yes

4. Have the authors made all data underlying the findings in their manuscript fully available (please refer to the Data Availability Statement at the start of the manuscript PDF file)?

Reviewer #1: Yes

5. Is the manuscript presented in an intelligible fashion and written in standard English?

Reviewer #1: Yes

6. Review Comments to the Author

Reviewer #1: (No Response)

7. PLOS authors have the option to publish the peer review history of their article (what does this mean?). If published, this will include your full peer review and any attached files.

**Do you want your identity to be public for this peer review?** For information about this choice, including consent withdrawal, please see our Privacy Policy.

Reviewer #1: No
